# 3D culture of functional human iPSC-derived hepatocytes using a core-shell microfiber

**Shogo Nagata**[1], **Fumisato Ozawa**[1], **Minghao Nie**[2], **Shoji Takeuchi**[1,2,3]*

**1** Institute of Industrial Science, The University of Tokyo, Tokyo, Japan, **2** Department of Mechano-Informatics, Graduate School of Information Science and Technology, The University of Tokyo, Tokyo, Japan, **3** International Research Center for Neurointelligence (IRCN), The University of Tokyo Institutes for Advanced Study (UTIAS), The University of Tokyo, Tokyo, Japan

* takeuchi@iis.u-tokyo.ac.jp

## Abstract

Human iPSC-derived hepatocytes hold great promise as a cell source for cell therapy and drug screening. However, the culture method for highly-quantified hepatocytes has not yet been established. Herein, we have developed an encapsulation and 3D cultivation method for iPSC-hepatocytes in core-shell hydrogel microfibers (a.k.a. cell fiber). In the fiber-shaped 3D microenvironment consisting of abundant extracellular matrix (ECM), the iPSC-hepatocytes exhibited many hepatic characteristics, including the albumin secretion, and the expression of the hepatic marker genes (*ALB*, *HNF4α*, *ASGPR1*, *CYP2C19*, and *CYP3A4*). Furthermore, we found that the fibers were mechanically stable and can be applicable to hepatocyte transplantation. Three days after transplantation of the microfibers into the abdominal cavity of immunodeficient mice, human albumin was detected in the peripheral blood of the transplanted mice. These results indicate that the iPSC-hepatocyte fibers are promising either as *in vitro* models for drug screening or as implantation grafts to treat liver failure.

## Introduction

Hepatocytes derived from human induced pluripotent stem cells (iPSC-hepatocytes) are promising cell sources in the fields of drug development, transplantation, and regenerative medicine [1]. For the culture of iPSC-hepatocytes, in comparison with the two-dimensional (2D) culture methods, three-dimensional (3D) culture methods have drawn much research attention recently; 3D microenvironments can promote the physiologically relevant hepatic functions of the hepatocytes [2,3]. For the 3D culture of hepatocytes, spheroid formation, in which dissociated hepatocytes are spontaneously aggregated by cell-cell interactions, is conventionally used [4,5]. However, there are problems associated with spheroid culture; cell aggregates are formed depending on their cell-cell junctions in suspension cultures which lack ECM. It is difficult to add the optimal amount and type of ECM to the 3D microenvironment in the conventional spheroid culture conditions. ECM is an important factor for the positive regulation of hepatocyte characteristics in various 3D hydrogel culture conditions [6,7]; the cell-ECM interaction promotes hepatic functions [8,9] and prevents cell death such as anoikis

**Data Availability Statement:** All relevant data are within the manuscript and its Supporting Information files.

**Funding:** S.N. is partly supported by JSPS KAKENHI (https://www.jsps.go.jp/j-grantsinaid/12_kiban/index.html, Grant Number 16H06329), and

the Japan Agency for Medical Research and Development (AMED), Research Center Network for Realization of Regenerative Medicine (https://www.amed.go.jp/program/list/01/02/001.html, 16bm0304005h0004 and 18bm0404021h0001).

**Competing interests:** ST is an inventor on intellectual property rights related to the cell fiber technology, and stockholders of Cellfiber Inc, a start-up company based on the cell fiber technology. However, any authors are NOT employed by the commercial company: Cellfiber Inc. Thus, this does not alter our adherence to PLOS ONE policies on sharing data and materials.

(loss of cell anchorage triggers apoptosis), which is induced in dissociated cells during the reconstruction of cell-cell and cell-ECM interactions in the suspension culture [10].

In this study, we establish the 3D culture of human iPSC-derived hepatocytes in Matrigel using a microfluidic fiber encapsulation technique called "cell fibers" [11]. The cell fiber is constructed using a 3D ECM-rich microenvironment as the core and mechanically stable alginate hydrogel as the shell. To construct the cell fibers based on iPSC-derived hepatocytes, we first mix commercially available human iPSC-hepatocytes with Matrigel and encapsulate the mixture into the core of the hydrogel microfibers; then the Matrigel is crosslinked and the cells are cultured in this 3D microenvironment. We demonstrate the advantage of our fiber culture conditions by examining the cell functions in the cell fibers both *in vitro* and *in vivo*. *In vitro*, we show that the hepatic function of hepatocytes can be enhanced using microfibers by assessing the secretion of albumin and hepatocyte-specific protein of the cell fibers, and by comparing the gene expression pattern of the hepatic cell markers, hepatic stem cell markers, and members of the CYP family among the fibers, spheroids, and 2D culture. *In vivo*, we show that the iPSC-hepatocyte fibers have good handleability and function as transplantable grafts by transplanting the iPSC-fibers into the abdominal cavity of mice.

## Materials and methods

### Cell culture

For conventional 2D culture, the commercially available iPSC-hepatocytes (REPROCELL) were seeded onto Matrigel-coated culture plate at a cell density of $2.4 \times 10^5$ cells/cm$^2$, following the manufacturer's instructions. For the spheroid culture, the iPSC-hepatocytes were suspended in their culture medium at a cell density of $1.0 \times 10^5$ cells/mL, and cell aggregates were formed by suspension culture in low-attachment 96-well V-bottom culture plates, a cell density of $1.0 \times 10^4$ cells/well.

### Formation and cultivation of core-shell hydrogel microfibers

Core-shell hydrogel fibers were prepared using a double-coaxial laminar-flow microfluidic device that was fabricated as described previously [11]. For the fiber formation, the iPSC-hepatocytes (REPROCELL) were recovered following the manufacturer's instructions and suspended in Matrigel (CORNING) as the core solution. A pre-gel solution of 2.0% Na-alginate (ALG100, Mochida Pharmaceutical Co. Ltd.) in saline was used as the shell solution, and CaCl$_2$ solution (100 mM CaCl$_2$, 3% sucrose) or BaCl$_2$ solution (20 mM BaCl$_2$, 250 mM D-Mannitol, and 25 mM HEPES) was used as the sheath solution for alginate gelation. The typical flow rates of the core, shell, and We compared the morphology respectively. The fibers generated in the device were collected in a tube filled with CaCl$_2$ (for *in vitro* assay) or BaCl$_2$ solution (for *in vivo* assay) and were incubated in the collection bath for 10 min. Then, the fibers were washed with DMEM medium (Sigma-Aldrich) to remove the sheath solution and transferred to a culture dish filled with ReproHepato Culture Medium (REPROCELL) for 3D cultivation under conditions of 5% CO$_2$ at 37˚C. The medium was changed 1, 3, and 5 days after starting the 3D cultivation.

### ELISA

For the measurement of human albumin *in vitro*, the supernatant of the culture medium was collected 3 and 6 days after the fiber formation and cultivation (n = 3, respectively). For the measurement of human albumin *in vivo*, peripheral blood was isolated from the transplanted

and sham mice (n = 3, respectively). A human albumin-specific ELISA kit (Bethyl Laboratories) was used according to the manufacturer's instructions.

## Immunocytochemistry

Cultured fibers were fixed with 4% paraformaldehyde in saline for 10 min at room temperature, washed with saline solution, and pre-treated with blocking buffer [5% BSA in PBST (PBS with 0.1% Triton X-100)] at 4˚C overnight. The cells were immunostained with primary antibodies (S1 Table) in the antibody buffer [1% BSA in PBST] overnight at 4˚C and incubated with secondary antibodies in the antibody buffer for 1 h. After being mounted within a Slow-Fade^TM antifade mountant (Thermo Fisher) containing Hoechst 33342 dye (Lonza), the samples were observed using a confocal laser fluorescent microscope (LSM780, Carl Zeiss).

## Gene expression analysis

Total RNA was isolated from iPSC-hepatocytes cultured in the conventional 2D condition and the cell fiber (n = 3, respectively) using the PureLink® RNA Mini Kit (Thermo Fisher) and was then treated with recombinant DNase I (Takara Bio) to eliminate genomic DNA contamination. For RT-PCR, the total RNA was used for the reverse transcription reaction with Superscript III™ reverse transcriptase (invitrogen), according to the manufacturer's instructions. PCR was performed using the TB Green Premix Ex-Taq^TM II system (Takara). *GAPDH* was used as an internal control for *ΔΔCt* quantitation method. The primer sequences are provided in S2 Table. For whole-transcriptome analysis, RNA sequencing was performed by Macrogen Inc. (Seoul, Korea). Briefly, library construction was performed using TruSeq RNA Sample Prep Kit v2 (Illumina) according to the manufacturer's instructions. The sequencing was performed by Illumina NovaSeq 6000 platform. Expression profile was calculated for each sample and transcript/gene as read count and fragment per kilobase of transcript per million mapped reads (FPKM). Furthermore, differentially expressed genes (DEG) analysis was performed on a comparison pair as requested using FPKM. For significant lists, gene-set enrichment analysis was performed based on gene ontology (GO) (http://geneontology.org/). The result of the associated gene and the enrichment test was summarized by GO ID. The significance of specific GO ID in the enrichment test with the DEG set was calculated by modified fisher's exact test.

## *In vivo* assay

For the functional evaluation of the iPSC-hepatocyte fibers *in vivo*, $1 \times 10^6$ cells were encapsulated into a core-shell microfiber formed using $Ba^{2+}$ as a crosslinker for the gelation of the alginate hydrogel. The cell fibers were cultured for 7 days before transplantation. After washing the cell fibers using saline solution, they were picked up and placed onto spatulas using tweezers. For the transplantation of the cell fibers into the abdominal cavity, celiotomy was performed on the eight-week-old male NOD/SCID mice (n = 3). The fibers were transplanted into their abdominal cavities using the spatulas, and the incision was closed using the nonabsorbable surgical sutures. Three days after the transplantation, peripheral blood samples were collected from the cervical veins of the mice and evaluated by ELISA. This study was carried out in strict accordance with the recommendations in the Guidelines for Proper Conduct of Animal Experiments of Science Council of Japan. The protocol was approved by the University of Tokyo Institutional Animal Care and Use Committee (Permission number: 26–14). For anesthesia and euthanasia, Isoflurane was used in this study, and all efforts were made to minimize suffering.

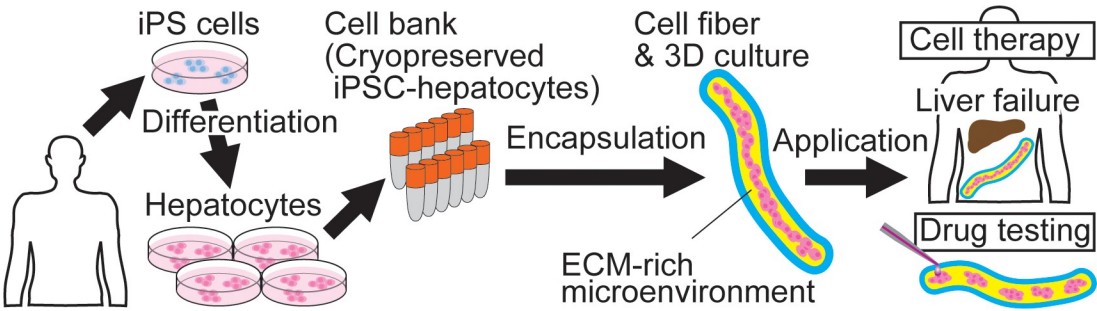

**Fig 1. Conceptual illustration of iPSC-hepatocyte fiber technology.** Human iPSC-hepatocytes are encapsulated in a 3D ECM-rich microenvironment using the cell fiber technology, and the iPSC-hepatocyte fibers are applicable to cell therapy for liver failure and drug development.

## Statistical analysis

Results are expressed as mean ± s.d. The data were analyzed using the two-tailed Student's t-test.

## Results and discussion

### Hydrogel encapsulation and 3D cultivation of human iPSC-hepatocytes in a cell-laden core-shell microfiber

We used commercially available human iPSC-hepatocytes (cryopreserved) as a cell source. To demonstrate that the core-shell microfiber culture condition is useful for the encapsulation and 3D culture of iPSC-hepatocytes (Fig 1), we compared this system with the conventional 2D and spheroid culture conditions. For the fiber encapsulation, the iPSC-hepatocytes were thawed and suspended in Matrigel, and the Matrigel solution containing the iPSC-hepatocytes was then used as the core solution for the formation of core-shell microfibers (S1A Fig). In order to mimic the physiologically relevant 3D microenvironment, we chose Matrigel as the ECM for the fibers; the Matrigel contains basal membrane components and is known to be able to support the expression of hepatocyte function [8]. The initial cell density in the core of the fibers was fixed to $1.0 \times 10^8$ cells/mL because the hepatocytes need not only cell-ECM interactions but also cell-cell interactions to survive and express their functions (Fig 2A).

We compared the morphology of the iPSC-hepatocytes cultured for a week in the conventional 2D, spheroid, and the cell fiber culture conditions. In the 2D culture, the hepatocytes attached to the Matrigel-coated plate, and formed 2D cell-cell interactions and exhibited a

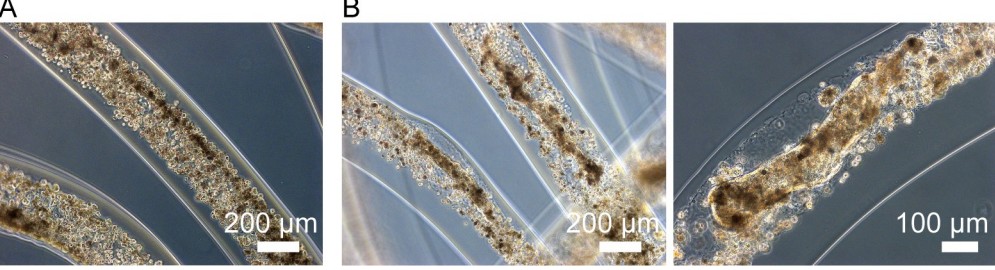

**Fig 2. Cell encapsulation and 3D cultivation of human iPSC-hepatocytes in a core-shell hydrogel fiber.** (A) Human iPSC-hepatocytes singly dissociated and diluted into Matrigel were encapsulated into the core region of the fiber at a density of $1 \times 10^8$ cells / mL, using the cell fiber technology. (B) The iPSC-hepatocytes were cultured for 7 days in the ECM-rich 3D microenvironment, and compacted cell aggregates were acquired in the fiber.

cobblestone morphology, which is characteristic of typical hepatocytes. In the spheroid culture, loose spheroid in distorted shapes was generated (S2A Fig). In contrast, in the cell fiber culture, the iPSC-hepatocytes formed compact cell aggregates with various 3D structures (Fig 2B). We found that the iPSC-hepatocytes were alive in the aggregates in the fiber, but the cells that did not form aggregates were dead (S2B Fig); the aggregates maintained their shape even after the degradation of the alginate shell was degraded using alginate lyase (S2C Fig). These results suggest that both cell-ECM and cell-cell interactions are needed for cell survival and the formation of 3D compacted cell aggregates in the fibers (S1B Fig).

## Hepatic characterization of iPSC-hepatocytes in the cell fibers compared to those in the 2D and spheroid culture

For the iPSC-hepatocytes in the cell fibers, we analyzed our samples by both immunostaining and ELISA. First, the iPSC-hepatocyte fibers on day 7 after the encapsulation were fixed and immunostained for functional mature hepatocyte marker proteins, albumin, asialoglycoprotein receptor 1 (ASGPR1) [12], and the hepatic cell marker HNF4α [13]. Cells comprising the aggregates in the fibers were highly positive for these protein markers (Fig 3A and S3 Fig), indicating that the encapsulated iPSC-hepatocytes maintained their hepatic characteristics during the 3D culture. The hepatic-marker-positive cells formed cell aggregates in the core region of the fibers. Second, using the ELISA, we evaluated whether the iPSC-hepatocytes were potent to secrete albumin *in vitro*. On both day 3 and day 6 of 3D culture, albumin protein was detected from culture medium at concentrations of 10.9 ± 3.8 ng/ml and 84.6 ± 37.2 ng/ml (mean ± s.d.), respectively (Fig 3B). These results indicate that the encapsulated iPSC-hepatocytes secreted albumin, expressed gradually enhanced hepatic characteristics with increasing culture time and that the shell of the fiber does not prevent the permeation of the secreted albumin protein into the culture medium. These results suggest that the iPSC-hepatocytes progress their maturation in the ECM-rich microenvironment of the fibers. Thus, the fiber culture can be useful for constructing functional 3D hepatic tissue *in vitro*.

We then compared the quality of the iPSC-hepatocytes in the cell fibers, spheroids, and 2D culture conditions. The expression level of the hepatic-specific genes *ALB*, and *HNF4α*, were evaluated by quantitative reverse transcription-polymerase chain reaction (qRT-PCR). The iPSC-hepatocytes in the fibers showed significantly high expression level of both *ALB* and *HNF4α*, compared to those in the spheroids and 2D culture conditions (Fig 3C). In addition, in our 2D culture, the hepatocytes exhibited a higher hepatic property than those in the spheroid culture, although it is generally known that 3D cultures—including spheroids—greatly enhance the cell characteristics of primary and stem cell-derived hepatocytes [2–9]. The purchased iPSC-hepatocytes used in this study were found to be unsuitable for the spheroid culture (S2A Fig); in literature, to form spheroids from the commercially available iPSC-hepatocytes, magnetic beads were used to modify the conventional suspension culture in order to support the efficient generation of cell-cell interactions [14]. In the cell fibers, the iPSC-hepatocytes were compacted to form cell aggregates, unlike those in the spheroids (Fig 2B), suggesting that the Matrigel functions as an ECM-rich microenvironment and supports the 3D culture of hepatocytes, with the upregulated expression of hepatic cell characteristics and prevented anoikis (S1B Fig).

## Comparison of gene expression patterns of the iPSC-hepatocytes in the fiber, spheroid, and 2D culture

To precisely evaluate the characteristics of the iPSC-hepatocytes cultured in the fibers, we performed qRT-PCR for the quantification of hepatic genes. Gene expression levels of the

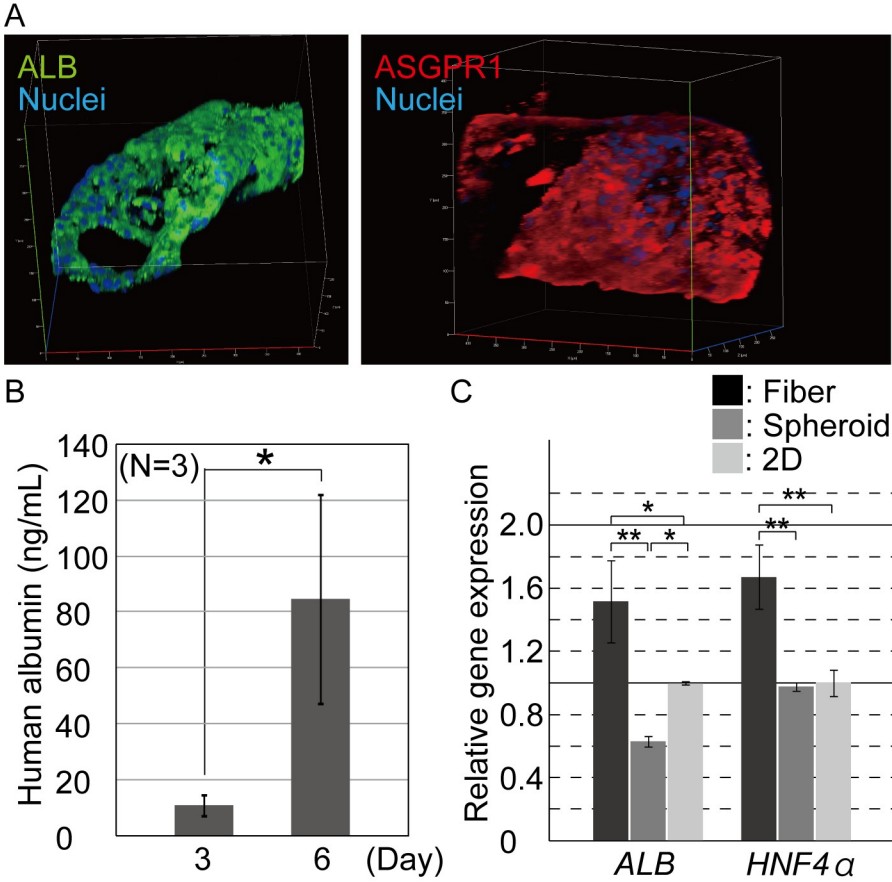

**Fig 3. Characterization of the hepatic function of the encapsulated hepatocytes.** (A) Immunocytochemistry was performed for hepatic marker proteins 7 days after hepatocyte cultivation. The iPSC-hepatocytes were positive for albumin and ASGPR1. (B) ELISA was performed for quantifying the albumin secreted into the culture medium. Secreted albumin was detected and its contents increased in the fibers during the culturing process (n = 3, $P < 0.07$). The error bars represent the standard deviation (s.d.) of triplicate samples. (C.) Quantitative RT-PCR was performed for the hepatic marker genes, *ALB*, and *HNF4α*. The expression of the marker genes in the hepatocytes from the cell fiber culture was significantly upregulated, compared to those from the conventional spheroid and 2D culture conditions (n = 3, *; $P<0.05$, **; $P<0.01$). The error bars represent the s.d. of triplicate samples.

hepatocytes in the fiber were compared to those in the 2D culture condition (the gold standard of hepatocyte culture). Compared to the iPSC-hepatocytes in the 2D culture, those in the cell fibers showed high expression of *ALB* and *HNF4α* (Fig 3C); the mature hepatocyte marker *ASGPR1*, early hepatic-lineage marker *TBX3*, the immature hepatocyte marker *AFP*, and the hepatic stem cell markers *EpCAM* [15] and *CPM* [16] were also significantly highly expressed in the fibers (Fig 4A). These results indicate that the iPSC-hepatocytes cultured in the fibers were "younger" hepatocytes than those in the 2D culture, and that the hepatic characteristics were promoted after their encapsulation into the ECM-rich 3D microenvironment and culturing using the core-shell fibers. In addition, the expressions of another marker of hepatic function, i.e. the CYP family genes which encode the CYP enzymes for the metabolism of chemicals, was also evaluated. Compared to the 2D culture, in our cell fiber culture, the iPSC-hepatocytes showed highly upregulated expression of the CYP family genes including *CYP2D6*, *CYP2C9*, *CYP2C19*, *CYP3A4*, and *CYP3A7* (Fig 4B), suggesting that drug metabolism capacity of the iPSC-hepatocytes in the fiber culture was higher than those in the 2D culture. To precisely investigate effects of the encapsulation of the iPSC-hepatocytes into the

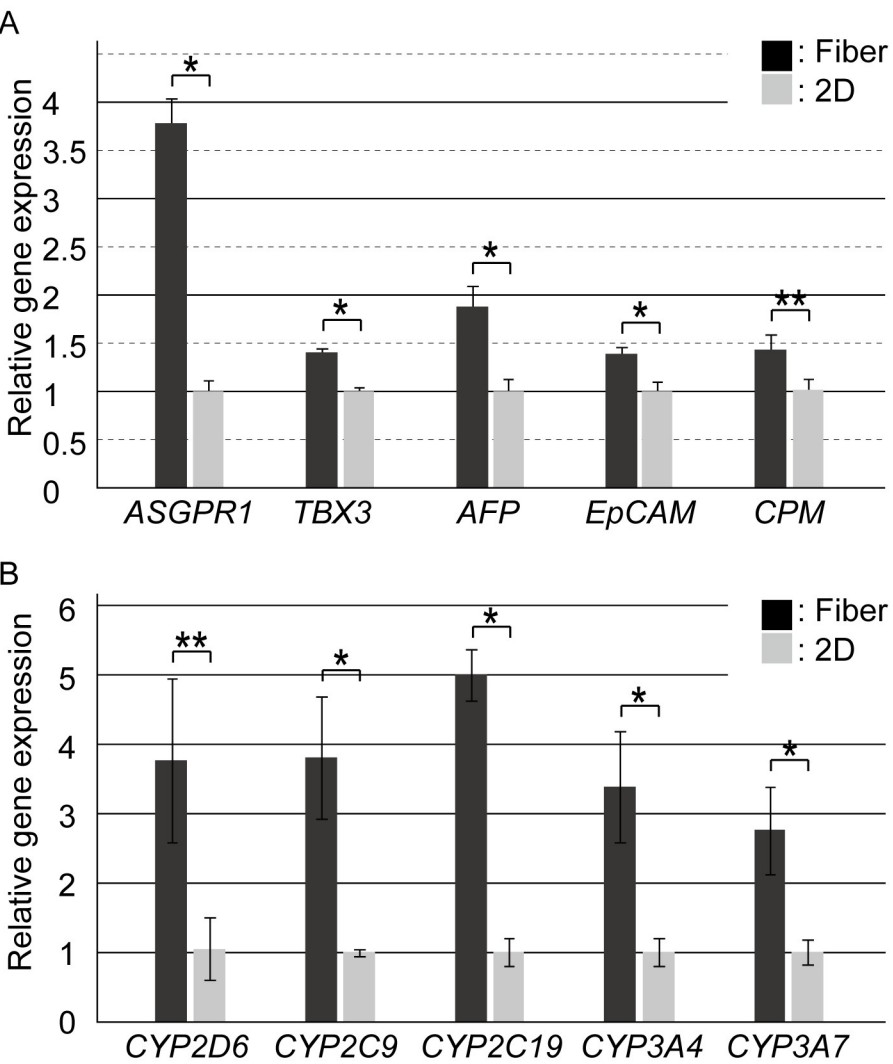

**Fig 4. Evaluation of the hepatic gene expression profile of the iPSC-hepatocytes cultured in the cell fibers.** (A) qRT-PCR was performed for marker genes of the developmental stage of hepatocytes (n = 3, *; $P<0.05$), **; $P<0.01$). The error bars represent the s.d. of triplicate samples. (B) qRT-PCR was performed for quantifying the CYP family genes, which encode the CYP enzymes for drug metabolism (n = 3, *; $P<0.05$, **; $P<0.01$). The error bars represent the s.d. of triplicate samples.

fiber, RNA-sequencing was performed, and the 2D and the cell fiber culture condition was compared on global gene expression. In S4A Fig, the heat map of the one-way hierarchical clustering (1,053 genes satisfying with fold change 2 (FC2)), and top 30 genes which were upregulated or downregulated in the cell fiber were shown based on DEG analysis, indicating that global gene expression of the iPSC-hepatocytes was changed in the cell fiber. In addition, gene ontology analysis indicated that not only hepatocyte-specific pathways but also wide biological processes were changed (S4B Fig). Moreover, to evaluate their hepatic developmental stage in the ECM-rich 3D microenvironment, the gene expression of hepatocyte-related integrin was assessed. The expression level of *ITGA1*, *ITGA5*, *ITGA6*, and *ITGB1* were also assessed since they are known to be highly expressed in hepatoblasts and fetal hepatocytes [17]. As a result, hepatocytes in the fibers and the 2D cultured were unchanged. The expression of *ITGB3*, which is associated with cellular senescence [18], was not also changed (S4C Fig),

indicating that the cell fiber culture did not affect the expression profile of the hepatocyte-related integrin genes.

Although the iPSC-hepatocytes are expected to be used for chemical screening of efficacy and toxicity for drug development, it has been known that the hepatocytes show low drug metabolism functions in conventional 2D culture condition; hence, various culture methods and conditions have been developed and investigated to promote the functions of cultured iPSC-hepatocytes [2–4,19,20]. Our results suggest that the iPSC-hepatocyte fibers can significantly promote hepatic function and therefore be one of the candidates to replace 2D cultures for drug screening.

## Application of the iPSC-hepatocyte cell fibers for cell transplantation

To evaluate the applicability of the iPSC-hepatocyte fibers as functional transplantable grafts, we performed an *in vivo* assay for albumin secretion after the transplantation of the microfibers into the abdominal cavity of immunodeficient NOD/SCID mice. For transplantation, our cell fibers have two advantages over hydrogel microbeads, which are widely used for cell encapsulation and transplantation research: first, the fiber shape is continuous in comparison to the separated droplets/beads, and therefore, the fibers can be easily picked up by tweezers and be efficiently implanted and retrieved as a whole; second, the fiber shell consists of alginate hydrogel, which makes the fibers mechanically strong to prevent the unwanted broad dispersion of the transplanted cells away from the injection site (Fig 5A). In the experiment, we transplanted the fibers that initially contained $1 \times 10^{6}$ cells into the abdominal cavity of immunodeficient NOD/SCID mice using a spatula (Fig 5B). Three days after the transplantation, we assessed the human albumin concentration in the blood plasma of the mice by ELISA. Human

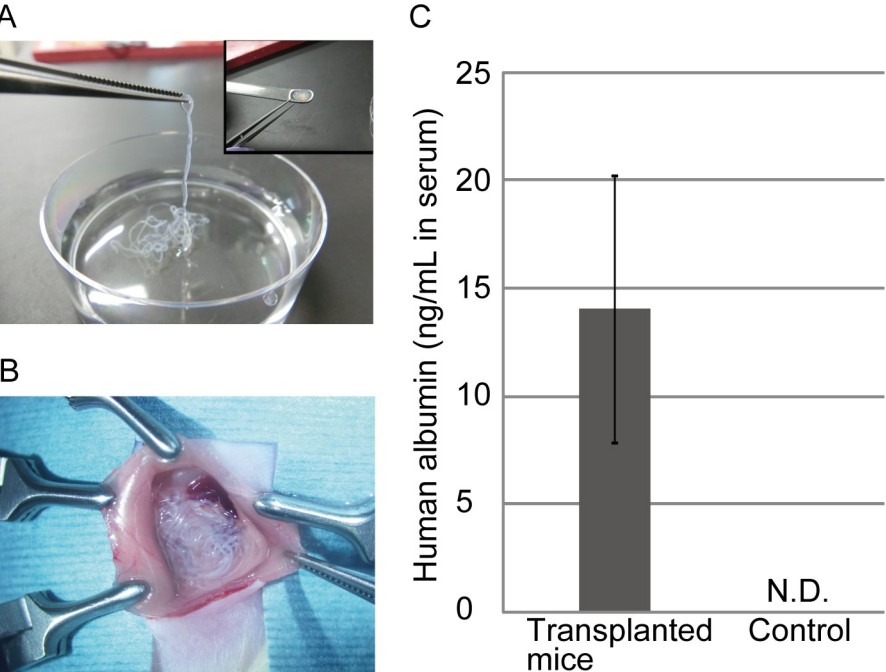

**Fig 5. Application of the iPSC-hepatocytes cell fibers as transplantable grafts.** (A) The fibers were easily picked up and collected in optimal shapes for transplantation. (B) The fibers were transplanted into the mouse abdominal cavity. (C) Human albumin was detected in the peripheral blood samples from the transplanted mice by ELISA (n = 3). The error bars represent the s.d. of triplicate samples. N.D.; not-detected.

albumin was detected at a concentration of 14.1 ± 6.2 ng/ml (mean ± s.d.) in the transplanted mice and was not detected in the blood of non-transplanted mice, which were used as controls (Fig 5C). The dosage of the secreted human albumin is enough to improve the survival ratio of acute liver failure (ALF) model mice as previously reported [21–23]. These results indicate that the cell fiber technology is useful not only for the 3D culture of human iPSC-hepatocytes with a high expression of hepatic functions but also for cell therapy, due to their mechanical strength and handleability as transplantable grafts.

In general, hepatocyte transplantation has been used as a symptomatic treatment for ALF, fulminant liver failure, and metabolic disorders to bridge the gap period when liver transplantation is not available due to the shortage of liver donors [24,25]. Furthermore, hepatocyte transplantation is expected to be applied especially for the treatment of congenital liver failure in the field of pediatric care [25,26]. Although there are many reports about clinical trials involving hepatocyte transplantation, these trials are not yet accepted as conventional treatment, compared to liver transplantation because the methods for hepatocyte preparation and transplantation are not yet well-established [27]. Therefore, the development and establishment of techniques for hepatocyte transplantation are highly desired. Our results suggest that transplantation of the iPSC-hepatocyte fibers can be a powerful tool for efficient and safe cell therapy for treating liver failure and also be an alternative option for enzyme replacement therapy.

## Conclusions

We developed a method for cell encapsulation and the 3D cultivation of human iPSC-hepatocytes in core-shell microfibers, with Matrigel as the ECM and alginate hydrogel. In the fibers, the iPSC-hepatocytes formed compacted cell aggregates, secreting albumin and highly expressing hepatic marker genes (including the CYP family genes). Furthermore, we demonstrate that the iPSC-hepatocyte fibers are mechanically stable and functional implantation grafts through the transplantation of the microfibers into the abdominal cavity of immunodeficient mice. In summary, the iPSC-hepatocyte fibers are suitable for the *in vitro* formation of 3D hepatic tissues, which can be applied for drug screening and cell therapy for liver disorders.

## Supporting information

**S1 Fig. Schematic illustration of the core-shell microfiber formation and cell culture in the ECM-rich 3D microenvironment.** (A) A double co-axial laminar flow microfluidic device was used for the formation of the iPSC-hepatocyte-laden core-shell hydrogel microfiber (iPSC-hepatocyte fiber). (B) The cells interacted with both other cells and the ECM, and were mechanically stimulated and regulated to spread and migrate three-dimensionally. Gradients of soluble factors, nutrients, and oxygen are also generated by diffusion through the ECM gel. (TIF)

**S2 Fig. 3D and 2D cultivation of human iPSC-hepatocytes.** (A) Human iPSC-hepatocytes formed loose and distorted spheroid in the suspension culture and exhibited a cobblestone morphology in the 2D culture on Matrigel-coated plate. (B) Live/dead staining was performed 7 days after the 3D culture of the hepatocyte fibers. The iPSC-hepatocytes in the cell aggregates were alive. (C) The shell of the fibers was degraded by alginate lyase after the 3D culture. The iPSC-hepatocytes maintained their compacted cell aggregates (arrowheads). (TIF)

**S3 Fig. Hepatic function characterization of the encapsulated hepatocytes 7 days after 3D cultivation.** Immunocytochemistry was performed for the hepatic stem/progenitor marker

EpCAM, and the hepatic marker HNF4α. Some of the iPSC-hepatocytes were positive for EpCAM merging with albumin, and almost of HNF4α-positive cells were also positive for ASGPR1.
(TIF)

**S4 Fig. Evaluation of the gene expression profile of the iPSC-hepatocytes in the cell fibers.** (A) RNA-sequencing was performed for hierarchical clustering analysis. Gene expression level is shown in normalized value (log2 based) using z-score, and color-coded with the color range shown in the top. Top 30 genes upregulated and downregulated in the fiber were selected and indicated. (B) Gene-set enrichment analysis which based on GO was conducted with the significant gene list and progressed about 3 categories of GO (biological processes, cellular component, and molecular function). The bar plot shown here is the top 10 terms of GO functional analysis in biological processes (*; P<0.05), **; P<0.01, ***; P<0.001). (C) qRT-PCR was performed for quantifying the expression of the integrin genes (n = 3). The error bars represent the s.d. of triplicate samples.
(TIF)

**S1 Table. Antibodies used for immunocytochemistry.**
(TIF)

**S2 Table. Primers used for qRT-PCR.**
(TIF)

## Acknowledgments

We thank Dr. T. Kido and Dr. A. Miyajima for their constructive comments and help with discussions regarding the experimental design. We also thank H. Aoyagi for assistance on the in-vivo assay, T. Saito for assistance with fiber formation, and Dr. T. Kobayashi for help with statistical analysis.

## Author Contributions

**Conceptualization:** Shogo Nagata, Shoji Takeuchi.

**Data curation:** Shogo Nagata.

**Formal analysis:** Shogo Nagata, Fumisato Ozawa, Minghao Nie.

**Investigation:** Shogo Nagata.

**Methodology:** Fumisato Ozawa.

**Visualization:** Shogo Nagata.

**Writing – original draft:** Shogo Nagata.

**Writing – review & editing:** Minghao Nie, Shoji Takeuchi.

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
