## [Decision Letter · Decision Letter 0]

31 Mar 2020

PONE-D-20-04950

3D culture of functional human iPSC-derived hepatocytes using a core-shell microfiber

PLOS ONE

Dear Prof. Takeuchi,

Thank you for submitting your manuscript to PLOS ONE. After careful consideration, we feel that it has merit but does not fully meet PLOS ONE’s publication criteria as it currently stands. Therefore, we invite you to submit a revised version of the manuscript that addresses the points raised during the review process.

We would appreciate receiving your revised manuscript by May 15 2020 11:59PM. To enhance the reproducibility of your results, we recommend that if applicable you deposit your laboratory protocols in protocols.io, where a protocol can be assigned its own identifier (DOI) such that it can be cited independently in the future. For instructions see: http://journals.plos.org/plosone/s/submission-guidelines#loc-laboratory-protocols

We look forward to receiving your revised manuscript.

Kind regards,

Hiroaki Onoe

Academic Editor

PLOS ONE

Journal Requirements:

"The authors declare competing financial interests: ST is a inventor on intellectual property rights related to the cell fibre technology, and stockholders of Cellfiber Inc, a start-up company based on the cell fibre technology."

We note that one or more of the authors are employed by a commercial company: Cellfiber Inc.

3. We note that you have a patent relating to material pertinent to this article. Please provide an amended statement of Competing Interests to declare this patent (with details including name and number), along with any other relevant declarations relating to employment, consultancy, patents, products in development or modified products etc. Please confirm that this does not alter your adherence to all PLOS ONE policies on sharing data and materials, as detailed online in our guide for authors http://journals.plos.org/plosone/s/competing-interests by including the following statement: "This does not alter our adherence to  PLOS ONE policies on sharing data and materials.” If there are restrictions on sharing of data and/or materials, please state these. Please note that we cannot proceed with consideration of your article until this information has been declared.

Reviewers' comments:

Reviewer's Responses to Questions

**Comments to the Author**

1. Is the manuscript technically sound, and do the data support the conclusions?

Reviewer #1: Yes

Reviewer #2: Partly

Reviewer #3: Yes

Reviewer #4: Yes

2. Has the statistical analysis been performed appropriately and rigorously? 

Reviewer #1: I Don't Know

Reviewer #2: Yes

Reviewer #3: Yes

Reviewer #4: Yes

3. Have the authors made all data underlying the findings in their manuscript fully available?

Reviewer #1: Yes

Reviewer #2: Yes

Reviewer #3: Yes

Reviewer #4: Yes

4. Is the manuscript presented in an intelligible fashion and written in standard English?

Reviewer #1: Yes

Reviewer #2: Yes

Reviewer #3: Yes

Reviewer #4: Yes

5. Review Comments to the Author

Reviewer #1: The authors reported a novel cultivation method of Human iPSC-hepatocytes with a fiber-shaped 3D scaffold. They encapsulated hepatocytes into the core of core-shell hydrogel fibers to achieve both a 3D ECM-rich environment and easy manipulation. The approach is interesting and they showed that the protein secretion and gene expression increase compare to 2D or spheroid culture. However, the results are a bit unclear because the manuscript lacks some information or experiments as follows:

1. For line 185 and Fig. 3B, there is a contradiction in the culture time until evaluating the albumin secretion (The manuscript is written as day 3 & 7, while the figure is shown as day 3 & 6). They need to be corrected.

2. In Fig. 3C, 4A, 4B, and S4, the authors compared the gene expression of hepatocytes. However, there is no description of the normalization of cell numbers. Since the number of cells affects to the amount of expression, the results should be normalized for comparison. The description should be added. If the results are not the normalized one, they should be revised.

3. In Fig. 5C and lines 276-280, the authors indicated that the concentration of albumin increases by transplantation of fibers compared to the non-transplanted case. However, since the purpose of the authors is to increase the cell function by their culture method, the effect should be compared not with the non-transplanted case but with the previous culture methods (e.g. spheroids).

Reviewer #2: Takeuchi and co-workers reported a culture system for human iPSC-derived hepatocyte-like cells using the core-shell microfibers. The authors compared the cell functions between the fiber and conventional 2D culture conditions, and claimed that the presented approach was effective. In addition, the authors perform in vivo experiments to show the applicability of the fiber to transplantation therapy. In general, organization of individual cells into 3D platforms is a good strategy to maximize the cell functions, and the presented paper is interesting as one of new approaches. However, following points should be properly explained and/or reflected before publication of this paper in this journal.

(1) It was unclear why the authors obtained good results when they encapsulated cells in the fiber, because there is no rational explanation for this point. There should be some important cell-cell and cell-matrix interactions in 3D formats that potentially enhanced cell functions, with proper signal transductions. These points, especially from the viewpoint of the molecular signalings, should be properly explained.

(2) The results of the animal study were not so convincing, because the authors did not mention the effectiveness of the fiber-based cell encapsulation and transplantation. Many researchers have transplanted cells (hepatocytes) that were encapsulated in hydrogel matrices, but the comparison with conventional strategies was not described at all in this paper. The reason for choosing the abdominal cavity as the transplantation site is also unclear; the liver itself, the kidney capsule, or subcutaneous site would be superior because of the presence of the blood flow. The blood concentration of human albumin might not be sufficiently high considering the number of the transplanted cells; I am wondering the cell viability. It was unclear what types of liver diseases could be improved by this approach, especially for humans. These points should be clarified.

(3) It was unclear why the authors used Matrigel as the matrix, instead of collagen, the gold-standard for hepatocyte culture.

(4) Some of the experimental conditions were not properly described. For example, following points are unclear: (i) the housekeeping gene used for qPCR, (ii) the age of the animal, and (iii) how many times the authors repeated the experiments, especially for the figures.

(5) The manuscript contains many typos and unnatural expressions. The entire manuscript should be thoroughly checked once again before submitting the revision.

Reviewer #3: This manuscript presents the 3D culture technique of human iPSC-derived hepatocytes using a core-shell microfiber. The core-shell microfiber is a unique, superior 3D culture platform that provides an ECM-abundant core and a mechanically-strong shell. The authors proved the potential of their original core-shell fibers for cell transplantation applications. The manuscript would seem of considerable interest to those working in tissue engineering and regenerative medicines. However, the authors should describe the differences with their previous publication in IEEE MEMS 2020 titled “3D hepatic tissue formed by iPSC-derived hepatocytes using a cell fiber technology.” Figure 1 includes the exactly same figures in Figure 3 of the previous publication. After polished based on the aforementioned critiques, this manuscript may be able to be published in PLOS ONE. I would recommend that this paper needs minor revision to be published in PLOS ONE.

Reviewer #4: Nagata et al. apply cell fiber biofabrication techniques to create core-shell fibers of human iPSC-derived hepatocytes and perform in vitro and in vivo characterizations, including transplantation in a mouse model. Overall, the work is thorough; however, there are several aspects that should be expanded before publication. First, the introduction is missing a lot of prior work on cell fibers. The state of the art should be discussed in more detail (e.g., which cells have been demonstrated as compatible with the cell fiber approach, why haven't human iPSC-derived hepatocytes been done before, etc.). Such discussion on how this work is different from prior developments is important. I think a figure and supplementary movie featuring the biofabrication process is needed. Fabrication results are included in Fig. 2, but that figure should be expanded with the fabrication setup and process, which will be helpful for the readership. Also, the conclusion reads like a quick summary, but conclusions should really be used to provide some deeper insights into the study. There are some other minor notes, like the format of randomly including the figure captions in the main text being difficult for the reviewers and some weird callouts to the suppl. figures (S2A Fig). Overall though, should the aforementioned changes be made, my recommendation is for the manuscript to be accepted.

6. PLOS authors have the option to publish the peer review history of their article (what does this mean?). If published, this will include your full peer review and any attached files.

Reviewer #1: No

Reviewer #2: No

Reviewer #3: No

Reviewer #4: No

---

## [Author Response · Author response to Decision Letter 0]

5 May 2020

Response to Reviewer# 1

Comment 1: For line 185 and Fig. 3B, there is a contradiction in the culture time until evaluating the albumin secretion (The manuscript is written as day 3 & 7, while the figure is shown as day 3 & 6). They need to be corrected.

Response 1:

We apologize for the incorrect information and have corrected the statement in the manuscript. 

Comment 2: In Fig. 3C, 4A, 4B, and S4, the authors compared the gene expression of hepatocytes. However, there is no description of the normalization of cell numbers. Since the number of cells affects to the amount of expression, the results should be normalized for comparison. The description should be added. If the results are not the normalized one, they should be revised.

Response 2: 

We apologize for causing this misunderstanding. For analyzing gene expression, we used RT-PCR technique. In the assay, the gene expression level of each sample can be normalized by the expression level of the housekeeping gene instead of cell number. Here we choose GAPDH, which is widely used as the housekeeping gene, for the normalization, so it is not necessary to mention cell number of the samples. We have revised the sentence on quantitative RT-PCR in the materials and methods section.

Comment 3: In Fig. 5C and lines 276-280, the authors indicated that the concentration of albumin increases by transplantation of fibers compared to the non-transplanted case. However, since the purpose of the authors is to increase the cell function by their culture method, the effect should be compared not with the non-transplanted case but with the previous culture methods (e.g. spheroids).

Response 3: 

We appreciate the reviewer for this comment. In this study, we demonstrate that the cell fiber technique improve hepatocyte functionality in vitro, and we have not compared the function of the fibers as a transplant with conventional hepatocyte transplantation methods in the animal experiment. Here, we focus on showing the possibility of applying the hepatocyte fibers to cell transplantation therapy. This is because the transplantation of the hepatocyte fibers into the abdominal cavity performed in this study is a completely novel transplantation method, and thus it is difficult to compare with the conventional transplantation methods, in which hepatocytes are transplanted into the liver by vein administration of hepatocytes.

Response to Reviewer# 2

Comment 1: It was unclear why the authors obtained good results when they encapsulated cells in the fiber, because there is no rational explanation for this point. There should be some important cell-cell and cell-matrix interactions in 3D formats that potentially enhanced cell functions, with proper signal transductions. These points, especially from the viewpoint of the molecular signalings, should be properly explained.

Response 1: 

We appreciate the reviewer for this comment. We are also focusing on the molecular mechanism on the enhancement of hepatocyte characteristics of the encapsulated iPSC-hepatocytes in the fiber. This is still challenging and many researchers are investigating the issue. Here we evaluated global gene expression profiles and compared the whole-transcriptome between the hepatocytes in the fiber and them in the 2D culture condition using RNA-sequencing technique. We have found some genes as candidates for enhancing hepatocyte characteristics in the fiber, but the details of their molecular functions are still under analysis and further research is needed in the future. In this revision, we do not mention the molecular mechanism but show the data of the RNA-sequencing analysis in S4 Fig. Furthermore, we have revised the sentences as shown below.

[Revised manuscript] (Results and Discussion section “To precisely investigate…” line 263)

To precisely investigate effects of the encapsulation of the iPSC-hepatocytes into the fiber, RNA-sequencing was performed, and the 2D and the cell fiber culture condition was compared on global gene expression. In S4A Fig, the heat map of the one-way hierarchical clustering (1,053 genes satisfying with fold change 2 (FC2)), and top 30 genes which were upregulated or downregulated in the cell fiber were shown based on DEG analysis, indicating that global gene expression of the iPSC-hepatocytes was changed in the cell fiber. In addition, gene ontology analysis indicated that not only hepatocyte-specific pathways but also wide biological processes were changed (S4B Fig).

Comment 2: The results of the animal study were not so convincing, because the authors did not mention the effectiveness of the fiber-based cell encapsulation and transplantation. Many researchers have transplanted cells (hepatocytes) that were encapsulated in hydrogel matrices, but the comparison with conventional strategies was not described at all in this paper. The reason for choosing the abdominal cavity as the transplantation site is also unclear; the liver itself, the kidney capsule, or subcutaneous site would be superior because of the presence of the blood flow. The blood concentration of human albumin might not be sufficiently high considering the number of the transplanted cells; I am wondering the cell viability. It was unclear what types of liver diseases could be improved by this approach, especially for humans. These points should be clarified.

Response 2: 

We appreciate the reviewer for this comment. As the reviewer pointed out, there are reports on transplantation of hepatocytes encapsulated in the hydrogel, but, in most of the studies, hydrogel microbeads are used for the cell encapsulation as we mentioned in the manuscript. The advantage of our cell fiber technology compared to the microbeads is that, by using a hydrogel microfiber having core-shall structure, the core retained the encapsulated hepatocytes in the ECM-rich 3D microenvironment, while the outer shell composed of alginate hydrogel supported the physical strength and stability in vivo of the cell fiber. Furthermore, since the fiber is possible to be handled as a single transplant due to its shape and strength, the transplant operation can be simple. In addition, when a malignant event occurs in the transplant or when it is no longer needed, the transplanted fiber can be more efficiently retrieved from the host than the microbeads.

 In this study, we chose intraperitoneal transplantation of the fiber because it is generally known that transplantation of hydrogel encapsulated hepatocytes into the peritoneal cavity is an attractive option for the management of acute liver failure providing short-term support to allow native liver regeneration [PLoS One. 2014 Dec 1;9(12):e113609.]. As the reviewer commented, transplantation into regions rich in blood vessels (e.g. subcutaneous site) is especially required for organ transplantation that requires permanent expression of cellular function, but, in this study, hepatocyte fiber transplantation is thought as one of novel cell therapy techniques and required to express the function in the short term as mentioned above. To clearly show that, we have revised the sentence as shown below.

[Revised manuscript] (Results and Discussion section “For transplantation,…” line 300)

For transplantation, our cell fibers have two advantages over hydrogel microbeads, which are widely used for cell encapsulation and transplantation research: first, the fiber shape is continuous in comparison to the separated droplets/beads, and therefore, the fibers can be easily picked up by tweezers and be efficiently implanted and retrieved as a whole; second, the fiber shell consists of alginate hydrogel, which makes the fibers mechanically strong to prevent the unwanted broad dispersion of the transplanted cells away from the injection site (Fig 5A).

Comment 3: It was unclear why the authors used Matrigel as the matrix, instead of collagen, the gold-standard for hepatocyte culture.

Response 3: 

We appreciate the reviewer for this comment. Type I collagen has been known as a suitable ECM for hepatocytes, but Matrigel is also known to be one of the most useful matrices for static, traditional hepatocyte culture [Tissue Eng Part A. 2010 Mar; 16(3): 1075–1082.]. Here we chose Matrigel for constructing the cell fiber due to basal membrane components for supporting the iPSC-hepatocytes in the core region of the fiber. Since various gels can be used for the cell fiber formation, it is possible to use an arbitrary gel (e.g. collagen) depending on the application.

Comment 4: Some of the experimental conditions were not properly described. For example, following points are unclear: (i) the housekeeping gene used for qPCR, (ii) the age of the animal, and (iii) how many times the authors repeated the experiments, especially for the figures.

Response 4: 

We appreciate the reviewer for this comment and have revised the sentence in the materials and methods section.

Comment 5: The manuscript contains many typos and unnatural expressions. The entire manuscript should be thoroughly checked once again before submitting the revision.

Response 5: 

We have checked again and revised the manuscript according to advice from experts on English.

Response to Reviewer# 3

Comment :This manuscript presents the 3D culture technique of human iPSC-derived hepatocytes using a core-shell microfiber. The core-shell microfiber is a unique, superior 3D culture platform that provides an ECM-abundant core and a mechanically-strong shell. The authors proved the potential of their original core-shell fibers for cell transplantation applications. The manuscript would seem of considerable interest to those working in tissue engineering and regenerative medicines. However, the authors should describe the differences with their previous publication in IEEE MEMS 2020 titled “3D hepatic tissue formed by iPSC-derived hepatocytes using a cell fiber technology.” Figure 1 includes the exactly same figures in Figure 3 of the previous publication. After polished based on the aforementioned critiques, this manuscript may be able to be published in PLOS ONE. I would recommend that this paper needs minor revision to be published in PLOS ONE.

Response :

We appreciate the reviewer for this comment. As pointed out by the reviewer, a related but preliminary work was reported in the MEMS conference. Comparing to the conference proceedings, we have significantly polished the initial ideas based on the molecular biological insight which was gained by new experiments that precisely evaluated the hepatic function and the cell characteristics and showed that the cell fiber was useful for efficient 3D culture system of the iPSC-hepatocytes and cell therapy for liver failure treatment.

On the comment for figure 1, we modified and revised parts of the figure.

Response to Reviewer# 4

Comment : Nagata et al. apply cell fiber biofabrication techniques to create core-shell fibers of human iPSC-derived hepatocytes and perform in vitro and in vivo characterizations, including transplantation in a mouse model. Overall, the work is thorough; however, there are several aspects that should be expanded before publication. First, the introduction is missing a lot of prior work on cell fibers. The state of the art should be discussed in more detail (e.g., which cells have been demonstrated ascompatible with the cell fiber approach, why haven't human iPSC-derived hepatocytes been done before, etc.). Such discussion on how this work is different from prior developments is important. I think a figure and supplementary movie featuring the biofabrication process is needed. Fabrication results are included in Fig.2, but that figure should be expanded with the fabrication setup and process, which will be helpful for the readership. Also, the conclusion reads like a quick summary, but conclusions should really be used to provide some deeper insights into the study. There are some other minor notes, like the format of randomly including the figure captions in the main text being difficult for the reviewers and some weird callouts to the suppl. figures (S2A Fig). Overall though, should the aforementioned changes be made, my recommendation is for the manuscript to be accepted.

Response :

We appreciate the reviewer for this comment. We considered adding a video of the fabrication process, but due to the limited activity of our research facility caused by the spread of the coronavirus infection (COVID-19), it is difficult to do it within the revise period. However, the biofabrication technique used in this study was basically the same as in the previous report [Nat Mater. 2013 Jun;12(6):584-90.], so the details can be omitted in this manuscript. The modification in the fiber fabrication is that, for the transplantation experiment, barium ions was used as the crosslinker of alginate gel for the fiber strength and in vivo stability. To clearly show that, we have revised the sentence as shown below.

[Revised manuscript] (Materials and methods section “The fibers generated in the …” line 91)

The fibers generated in the device were collected in a tube filled with CaCl2 (for in vitro assay) or BaCl2 solution (for in vivo assay), and were incubated in the collection bath for 10 min.

As the reviewer pointed out, there have been reports on the cell fibers. The cell fiber technology is thought to be applicable to all cell types in principle, but previous reports mainly focused on the fabrication technology using various cell lines, which are relatively stable in cultivation. The difference between them and this study is that here we developed the fibers using the iPSC-derived functional cells, which can be directly applied to medical applications (e.g. cell therapy). This is the first report of the fiber composed of iPSC-hepatocytes, which required to preciously evaluate the cell characteristics and transplantation experiments as performed in this study.

The paper format follows the PloS ONE submission rules. Therefore, the figure captions are inserted in the main text immediately after the paragraph that cites the figure. 

We appreciate the comment on S2A Fig and have revised the sentence as shown below.

[Revised manuscript] (Results and Discussion section “We compared the…” line 183)

We compared the morphology of the iPSC-hepatocytes cultured for a week in the conventional 2D, spheroid, and the cell fiber culture conditions. In the 2D culture, the hepatocytes attached onto the Matrigel-coated plate, and formed 2D cell-cell interactions and exhibited a cobblestone morphology, which is characteristic of typical hepatocytes. In the spheroid culture, loose spheroid in distorted shapes was generated (S2A Fig). In contrast, in the cell fiber culture, the iPSC-hepatocytes formed compact cell aggregates with various 3D structures (Fig 2B).

---

## [Decision Letter · Decision Letter 1]

27 May 2020

3D culture of functional human iPSC-derived hepatocytes using a core-shell microfiber

PONE-D-20-04950R1

Dear Dr. Takeuchi,

We are pleased to inform you that your manuscript has been judged scientifically suitable for publication and will be formally accepted for publication once it complies with all outstanding technical requirements.

With kind regards,

Hiroaki Onoe

Academic Editor

PLOS ONE

Additional Editor Comments (optional):

Reviewers' comments:

Reviewer's Responses to Questions

**Comments to the Author**

1. If the authors have adequately addressed your comments raised in a previous round of review and you feel that this manuscript is now acceptable for publication, you may indicate that here to bypass the “Comments to the Author” section, enter your conflict of interest statement in the “Confidential to Editor” section, and submit your "Accept" recommendation.

Reviewer #1: All comments have been addressed

Reviewer #2: All comments have been addressed

Reviewer #3: All comments have been addressed

Reviewer #4: All comments have been addressed

2. Is the manuscript technically sound, and do the data support the conclusions?

Reviewer #1: Yes

Reviewer #2: Yes

Reviewer #3: Yes

Reviewer #4: (No Response)

3. Has the statistical analysis been performed appropriately and rigorously? 

Reviewer #1: Yes

Reviewer #2: Yes

Reviewer #3: Yes

Reviewer #4: (No Response)

4. Have the authors made all data underlying the findings in their manuscript fully available?

Reviewer #1: Yes

Reviewer #2: Yes

Reviewer #3: Yes

Reviewer #4: (No Response)

5. Is the manuscript presented in an intelligible fashion and written in standard English?

Reviewer #1: Yes

Reviewer #2: Yes

Reviewer #3: Yes

Reviewer #4: (No Response)

6. Review Comments to the Author

Reviewer #1: Authors have properly addressed my concerns in the revision, and I agree to their explanations.

I have no more concerns on this article.

Reviewer #2: The revised manuscript by Takeuchi and co-workers properly reflected most of my previous concerns, and now the paper was improved. This paper is a nice example of the hydrogel fiber-based cell culture technique, and hence, I recommned accepting this paper for publication.

Reviewer #3: The authors replied to the reviewers' comments well and improved the manuscript. I recommend the manuscript for publication.

Reviewer #4: (No Response)

7. PLOS authors have the option to publish the peer review history of their article (what does this mean?). If published, this will include your full peer review and any attached files.

Reviewer #1: No

Reviewer #2: No

Reviewer #3: No

Reviewer #4: No

---

## [Editor Report · Acceptance letter]

2 Jun 2020

PONE-D-20-04950R1 

3D culture of functional human iPSC-derived hepatocytes using a core-shell microfiber 

Dear Dr. Takeuchi:

I'm pleased to inform you that your manuscript has been deemed suitable for publication in PLOS ONE. Congratulations! Your manuscript is now with our production department. 

Kind regards, 

on behalf of

Dr. Hiroaki Onoe 

Academic Editor

PLOS ONE